# A qualitative exploration of consumers' views and experiences toward online pharmacies: Narrative from a developing country

**Amal K. Suleiman**[1*], **Abbas Albarq**[2], **Ateeq Ur Rehman**[3]

**1** Department of Pharmacy Practice, College of Clinical Pharmacy, King Faisal University, Al Ahsa, Saudi Arabia, **2** Department of Management, School of Business, King Faisal University, Al Ahsa, Saudi Arabia, **3** Department of Engineering Technology, Foundation University Islamabad (FUSST), Islamabad, Pakistan

* aksuleiman@kfu.edu.sa, albarqok@yahoo.com

## Abstract

### Introduction

The widespread adoption of the internet and online pharmacies has made accessing medications more convenient, especially with the growing use of smartphones in developing countries. While these new purchasing methods offer ease of access, they also raise concerns about the prevalence of counterfeit drugs. This study examines consumer experiences with online pharmacies, focusing on the factors that influence purchasing decisions and the concerns associated with buying medicines and supplements in the context of a developing country.

### Method

An exploratory qualitative study was conducted in Islamabad/Rawalpindi (twin cities in Pakistan's federal area) from June to October 2023. Four focus group discussions (FGDs) were conducted, with each lasting approximately 30–40 minutes. The participants were approached in person, and the purpose of the study was explained to them. The participants were recruited after written informed consent was obtained. Grounded theory was used as the guiding methodology to explore consumer behavior toward products obtained from online pharmacies. After the culmination of data collection, the audio recordings originating from the FGDs were transcribed. Since the FGDs were conducted in Urdu, the audio recording was transcribed in Urdu verbatim. Subsequently, the transcripts were translated into English by an independent researcher who was fluent in Urdu and English. To ensure the accuracy, comprehensiveness, and impartiality of the transcriptions, independent verification of the transcripts was conducted by a second independent researcher.

**Data availability statement:** All relevant data are within the paper.

**Funding:** This work was supported by the Deanship of Scientific Research, Vice Presidency for Graduate Studies and Scientific Research, King Faisal University, Saudi Arabia [Grant No: KFU252892].

**Competing interests:** The authors have declared that no competing interests exist.

## Results

A total of thirty-two participants took part in the focus group discussions. The FGDs identified three themes, namely: reasons for the purchase of online medicines/supplements, psychological factors involved in the purchase of online medications and supplements, and concerns regarding online purchase of medicine and supplements. The themes were further divided into twelve subthemes. Participants cited the ease of payment, ease and convenience, low cost, choice, accessibility, and discounts as reasons for favoring online purchase of medicines and health/food supplements. Doubts about the efficacy of the medicines, safety concerns, and uncertainty regarding storage conditions were the psychological factors that study participants mentioned. Personal information leakage and the influx of unwanted advertisements via email, which they attributed to inadequate online security measures, were the main concerns regarding online purchase of medicine and supplements.

## Conclusion

This study offers new insights into consumer behaviors and concerns regarding online pharmacies in a developing country, adding to the limited literature on this topic. While online pharmacies offer convenience and can stimulate economic growth, the study emphasizes the need for stricter regulation to address consumer concerns and ensure safe and reliable purchasing experiences.

## Introduction

The internet has transformed various aspects of our lives, influencing communication as well as procurement practices and strategies [1]. As internet access continues to expand, its use for seeking health-related information has also increased [2]. Online pharmacies have been a developing channel of the pharmaceutical supply chain since the beginning of the century [3]. Online pharmacies have been growing for decades in the Western world, and in recent times, they have picked up pace globally, with the coronavirus (COVID-19) pandemic acting as a major catalyst [4]. In the wake of the pandemic, the shift toward online pharmacies has become even more prominent. The long-term impact of this shift has been a sustained increase in the sale of medications via online platforms [3,5,6]. A study from Hungary showed that the purchase of medicines using the internet increased from 4.17% in 2018 to 44.25% by March 2020 [2].

Due to advances in technology and easy access to the internet, businesses across the globe are swiftly embracing online shopping, leveraging the full array of advantages that online marketplaces provide [7]. Presently, the global valuation of the e-pharmacy (online pharmacy) market stands at approximately U.S.$81.6 billion, with projections indicating a growth trajectory to reach U.S.$244 billion by 2027 [4]. The utilization of the internet [8] and the widespread use of mobile devices [9] have significantly contributed to the surge in the acquisition of health products through online pharmacies and websites [10]. In addition to the convenient availability of medications through online pharmacies, customers have reported high satisfaction with pricing [11].

The online sale of medicines is associated with both public health concerns, such as the unauthorized sale of prescription-only medications and the dissemination of substandard and falsified drugs, and cybersecurity issues, encompassing consumer fraud and inadequate data privacy [12,13]. But online pharmacies present potential avenues for improving access to medications. As internet and smartphone penetration continues to expand, e-pharmacy holds the potential to enhance access for individuals with disabilities, the elderly, and those residing in rural areas [14]. The appeal of online pharmacies lies in the effortless availability of a diverse range of medicines and supplements that can be freely obtained through internet transactions [2]. Key motivating factors driving the growing inclination toward purchasing medicines and supplements online include convenience, cost-effectiveness, and broader product availability compared to traditional community pharmacies [15–19]. Furthermore, the user-friendly nature of online pharmacies, coupled with the prompt doorstep delivery of medications, proves particularly advantageous for older adults and those with physical disabilities who require regular access to medications for chronic illnesses [20].

Online pharmacies face issues and obstacles, one of which is the unregulated use of antibacterial drugs. Findings from a study in China revealed that 79% of online pharmacies dispensed antibiotics without a valid prescription [21]. Additionally, certain medications are illicitly sold without the requisite prescriptions [22]. Moreover, the unsupervised and self-administered use of drugs poses a significant risk to consumers, exposing them to harmful adverse effects and elevating the potential for morbidity and mortality [23]. The acquisition of medicines outside the conventional supply chain introduces various patient safety risks, notably the prevalence of counterfeit medications. Globally, the estimated proportion of counterfeit medicines is 10% [24], ranging from less than 1% in developed countries [25] to over 30% in developing countries such as in Africa, Asia, and Latin America [26]. Another concern associated with online pharmacy transactions is the absence of restrictions on customer age, allowing individuals under 18 to purchase medicines online, which is prohibited [27].

As a developing country and a low-middle-income nation, Pakistan is witnessing a rising trend in the adoption of online pharmacies. These platforms are gradually gaining traction and popularity as online businesses strive to establish trust and build rapport with consumers [28]. In Pakistan, the platforms for online shopping for medicines includes Dawaai. pk, DVAGO, and Tabiyat.pk [29]. Pakistan's internet-based pharmacy business is expected to develop at a compound annual growth rate of 16.8% between 2020 and 2025 [28]. The health infrastructure in Pakistan is inadequate and underdeveloped for chronic diseases, and the emergence of e-pharmacies/online pharmacies serves as a potential solution through access to medicine [29]. In 2012, the autonomous Drug Regulatory Authority of Pakistan (DRAP) was formed to provide effective pharmaceutical regulation in the country [30]. However, the role of DRAP in ensuring the quality and safety of medications and the regulatory framework for online pharmacies is still in its nascent stages [29]. Despite the growing popularity of online pharmacies in Pakistan, there is limited research on consumer attitudes toward e-pharmacies and the purchase of medication from online sources. Existing studies in other countries have highlighted factors such as convenience, cost-effectiveness, and product variety as key drivers of online pharmacy usage, but trust remains a major barrier for consumers. This study aims to address this gap by exploring customers' experiences and attitudes toward online pharmacies in Pakistan, specifically focusing on the factors influencing their purchasing decisions and the concerns they have regarding the safety and reliability of these platforms. As there was no psychometric tool available to measure these aspects, and the purpose was to gain an in-depth understanding rather than quantitative measurement, a qualitative approach was chosen as the most appropriate research method.

## Methods

### Operational definitions

**Online pharmacy/internet pharmacy.** In the current study, an online pharmacy (also known as an internet pharmacy, mail-order pharmacy or e-pharmacy) refers to internet retailers that sell non-prescription and prescription medicines directly to patients and provide information on their products and services via the internet [31].

## Drugs

As defined by the U.S. Food and Drug Administration, drugs are substances recognized by an official pharmacopoeia and are intended for use in the diagnosis, cure, mitigation, treatment, or prevention of disease [32].

## Counterfeit drugs

Counterfeit drugs are the products that are deliberately and fraudulently mislabeled with respect to their identity and/or source. The definition is applicable to both branded and generic products [33].

## Substandard drugs

According to the World Health Organization, substandard drugs do not meet quality standards and specifications, often because of poor manufacturing practices or inadequate quality control [34].

## Falsified drugs

Falsified medical products deliberately misrepresent their identity, composition or source. These products are often created and distributed with the intent to deceive consumers for financial gain [34].

## Health products

According to the Drug Regulatory Authority of Pakistan, health products are the products covered in enlistment rules, including food supplements and nutritional products and dietetic foods for infants [35].

## Short items

Short items are those medical items for which there is an inadequate supply, an imbalance between the demand and supply or production, or unavailability for a particular time period [36].

## Research design

The scarcity of evidence-based information regarding consumer behavior toward online pharmacies in Pakistan led to the adoption of a qualitative study design. Qualitative methods are ideal for inductive approaches aimed at generating hypotheses and have far more potential for health education research than any other model [37].

A qualitative methodology through focus group discussions (FGDs) was adopted. The reasons for selecting FGDs were multifactorial. Compared to other qualitative methods, FGDs are flexible and capable of in-depth exploration of respondents' attitudes and experiences [38,39]. Furthermore, FGDs have the unique ability to explore supportive ideologies and the differences of perspective between groups [40]. Lastly, FGDs not only examine how knowledge and ideas both develop and operate within a cultural context, but also fill the gaps that are often exposed by quantitative methods [39].

## Study settings and sampling

This study was carried out from June to October 2023. Community dwellers of the twin cities (Islamabad and Rawalpindi) in the federal area of Pakistan were approached for the FGDs. The study was conducted in a rented hall of a local hotel with facilities available for audio recording. A convenience sampling method was used to select participants from different backgrounds, including a variety of age, gender, education level, and health status, to get more details and experiences with online purchases. The required sample was determined by reaching the saturation point when no more new ideas could be obtained.

### Inclusion and exclusion criteria

Pakistani nationals of both genders and aged 18 years and above were eligible for this study. Furthermore, the respondents needed to have acquired either health or pharmaceutical products from an online source in the past two years. Those with no experience of online purchases, with mental health issues, and/or those not consenting to participate were excluded.

### Research tool (the guide, validation, reliability, and pilot study)

While this was forerunner research in Pakistan, an extensive literature review based on previous studies was conducted [2,3,10,18,41,42]. Rather than adopting these survey instruments directly, the study developed an interview guide based on the available evidence [43–46] and through expert panel discussion, and experience sharing [47–49]. Nevertheless, it was made sure that the attributes and attitudes of the local context were kept in mind and the guide was piloted before the actual study. The guide was established with widely framed, open-ended questions that gave enough opportunities to the respondents. It was also assured that a productive discussion about consumer experiences was carried out and factors that guide their preference for online purchases were particularly highlighted.

The guide was written in English and translated into Urdu (the national language of Pakistan) by an independent linguistic expert. The translated guide was back-translated into English to avoid discrepancies by another independent translator [50]. With a few amendments in translation, the guide was subjected to face and content validity by a panel of experts (senior researchers and marketing experts). Once the validity was ensured, the guide was piloted with five community dwellers to ensure that topics to be discussed were at the level that respondents would comprehend with ease. After this reliability test, the guide was made available for the main study. Data and participants of the pilot study were not included in the final analysis.

### The focus groups (composition and constitution)

Four discussion groups were conducted, with each FGD lasting approximately 30–40 minutes. Each group comprised six males and two females which provided diversified experiences and discussion among the participants.

### The focus group discussions (session and procedures)

The participants were approached in person and the purpose of the study was explained to them. The participants were recruited after written informed consent was obtained. The last author (PhD degree) acted as moderator and the FGDs were audio-taped with the consent of the participants. Keeping the nature of the study and the ease of the respondents in consideration, the interviews were conducted in Urdu. All participants were briefed about the study objectives before each FGD, and a debriefing session was conducted at the end of the discussion. To draw in-depth views, participants were given the freedom to add additional reviews and comments. The author was assisted by a research assistant who acted as an observer and assisted in monitoring the field notes, facial expressions, and body language that complemented the audio recordings.

### Data analysis and interpretation

Grounded theory was used as the guiding methodology to explore consumer behavior toward products obtained from online pharmacies [51]. All responses were analyzed using standard qualitative research techniques [52]. After the culmination of data collection, the audio recordings originating from the FGDs were transcribed. Since the FGDs were conducted in Urdu, the audio recording was transcribed in Urdu verbatim. Subsequently, the transcripts were translated by an independent researcher who was fluent in Urdu and English. To ensure the accuracy, comprehensiveness, and impartiality of the transcriptions, independent verification of the transcripts was conducted by a second independent researcher.

As per protocols of grounded theory, the analysis followed an iterative coding process, where open coding was initially employed to break the data into meaningful units [51]. Preliminary codes were developed by carefully reading the transcribed text and identifying key concepts. The subsequent phase involved focused coding, where the initial codes were refined, grouped, and categorized to form broader themes related to consumer attitudes and experiences with online pharmacies for health supplements and pharmaceutical products. Subsequently, the axial coding approach was used to analyze the sequences and to develop the relationship network for these codes, themes, and categories to create models for the decision-making process. To ensure the robustness and rigor of the coding process, two independent researchers were asked to review the codes and achieved an inter-coder reliability rate of 95%.

The sessions were conducted until thematic saturation was reached [53,54]. The research team analyzed the recordings (verbatim) and later arranged an informal gathering where participants were presented with the finalized scripts [55]. They were asked for confirmation of precision and accuracy of words, ideas, and jargon used during the script analysis. Once confirmed, the transcripts were translated into English by another independent translator for thematic content analysis [56,57]. All emerging themes and subthemes were discussed among the research team for accuracy and were presented for data extrapolation and interpretation.

### Ethical approval

The Institutional Ethical Committee of the Foundation University, Rawalpindi Campus, Pakistan approved the study (EC/FUI/2023/32). Written consent for participation and publication was taken from the respondents before the FGDs. The participants were introduced to the nature of the research before the beginning of the sessions and were assured of the confidentiality of their responses and their right to withdraw from the study.

### Results

*Demographic of participants.* Thirty-two participants were divided into four FGDs. There were twenty-four males and eight females. Nineteen participants had a bachelor's degree. Fifteen participants were in the age group of 29–38 years old.

Data from the FGDs revealed three major themes, namely reasons for the purchase of online medicines/supplements, psychological factors involved in the purchase of online medications and supplements, and concerns regarding online purchase of medicine and supplements. The themes were further divided into 12 subthemes, as shown in Fig 1.

**Theme 1: Reasons for the purchase of online medicines/supplements.** On a general note, participants positively perceived the online purchases of products. Accessibility, lower cost with more choice, discounts, and availability of short items were reasons for favoring the online purchase of medicines and health/food supplements. Most participants reported purchasing online products including over-the-counter drugs, multivitamins, herbal supplements, and other dietary supplements. However, a small proportion also reported acquiring medicines for chronic diseases. To extract comprehensive views, four subthemes were generated, which are discussed below.

*Subtheme 1(a): Accessibility and ease of purchasing online.* As expected, accessibility was the main reason for preferring an online venue. Although the participants reported varied reasons for their tendency toward online purchases, collectively the primary motivation behind selection was flexibility to shop at any time, from anywhere, and avoiding crowds and sales pressure.

*"Medicines, supplements and other related products are available online. Online purchase avoids long queues, no parking hassle within busy shopping hubs and markets and hence saves time and additional efforts of visiting a pharmacy". (P3 of FGD2; P6 of FGD 1)*

Continuing with accessibility and ease of purchase, delivery to the home was another reason for preferring online purchases. Where doorstep delivery allows customers to receive items without leaving their homes, it expands market reach

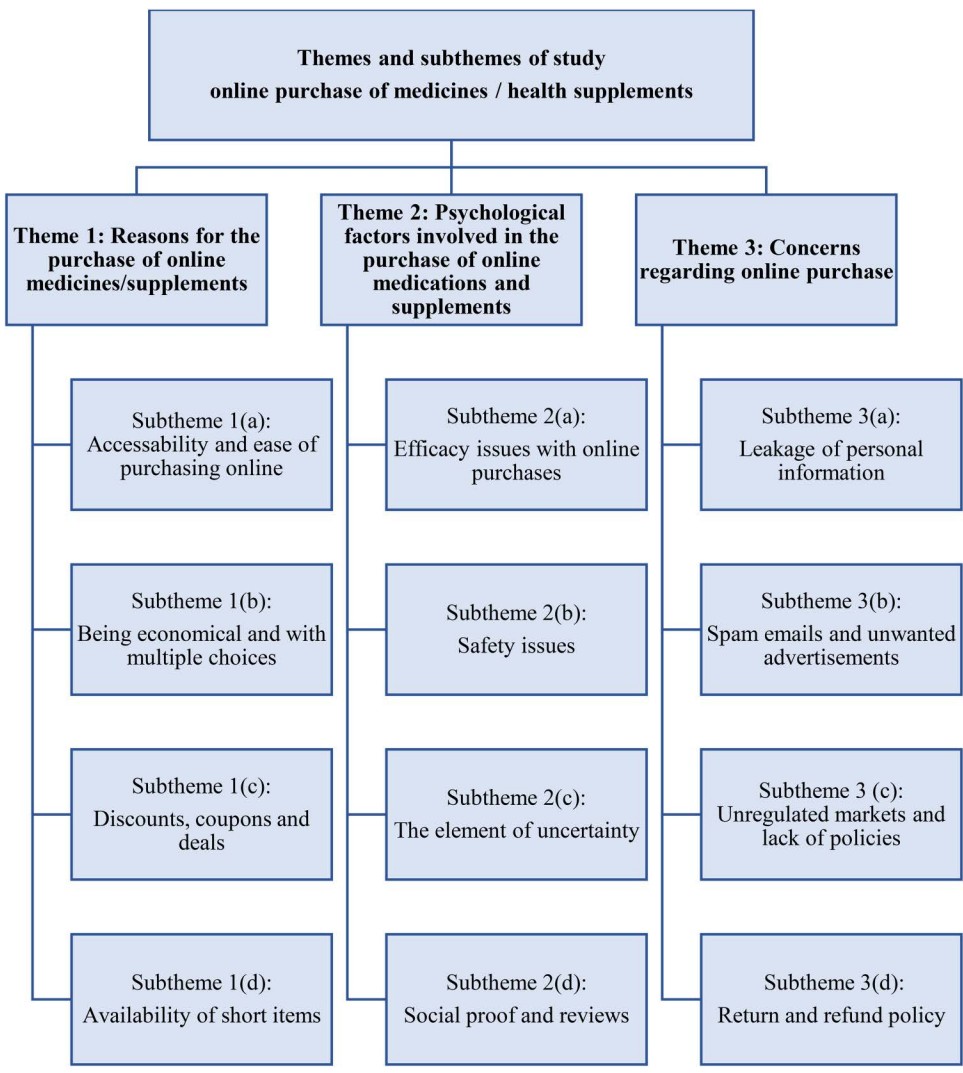

**Fig 1. Themes and subthemes derived from the study.**

and enhances customer satisfaction through faster delivery times and convenient scheduling options for businesspeople. This was also mentioned clearly by the members of the FGDs.

> *"The items or products are made available at doorstep once ordered from online pharmacies/websites. It saves a lot of time as well as give room for comfort as well while coming home from a busy and hectic day". (P5 of FGD3; P4 of FGD1)*

Finally, the alternative methods of payments available for purchases were another reason for selecting an online source. With options like digital wallets, mobile payments, and even cash on delivery, complex banking procedures are not required, and financial transactions are more convenient, seamless, and secure.

> *"Among the payment methods, mobile and wallets are a blessing. Now there is no need to carry cash every time and to pay for the items/medicines. It is easy, convenient and secure." (P4 of FGD1; P2 of FGD3)*

**Subtheme 1(b): Being economical and with multiple choices.**    It is known and accepted that online purchases are more economical than traditional in-store shopping. The reduced overhead costs give enough space to online retailers to offer reduced prices. The same reasons were also mentioned by the study respondents. Availability of multiple choices, being able to compare product specifications and going through reviews of prior buyers were rated as key determinants in preferring online sources to a physical purchase.

*"In addition to the specific brand, generic items are also available at low cost. As a customer, I have enough time to compare products and decide accordingly."* (P2 of FGD3; P8 of FGD3)

*"One of the participants (P5 of FGD3) further explained that "There are option and choices. I can select different products with the lowest price. Furthermore, the reviews by the buyers and end users also helps in purchasing a product".* (P5 of FGD3)

**Subtheme 1(c): Discounts, coupons and deals.**    Discounts, coupons, and deals are always lucrative for customers. Parallel to this, such offers attract new customers for online entrepreneurs. Discounts and deals are known to drive sales, increase customer retention and hence result in potential savings on purchases. Participants in this current study also mentioned that discounts and vouchers are attractive and for that reason they prefer online pharmacy products and supplements.

*"Online purchases from different websites have a competitive environment and they offer different discounts in terms of paying less money than the actual. In some cases, when purchasing two items from the same website/online pharmacy, they offer some item free as well. It saves money and this is not available at community pharmacies."* (P3 of FGD 1; P1 of FGD 2; P6 of FGD 4)

**Subtheme 1(d): Availability of short items.**    The participants mentioned that searching for short items through online sources is much easier than visiting different community pharmacies/retail pharmacies in person. Those items that are out of stock on the traditional market can be easily accessed while ordering/purchasing from these online pharmacies/websites.

*"Short items can be easily searched online. I must visit a lot of pharmacies to find a short item. It is frustrating and annoying to visit different venues in search of specific items which is not the case with online purchases."* (P5 of FGD 2; P3 of FGD 3)

***Theme 2: Psychological factors involved in the purchase of online medications and supplements.*** Despite positive experiences of engaging in online purchases, several psychological factors related to online purchases were highlighted during the analysis. The psychological factors involved in the purchase of online medications and supplements are summarized below.

A few participants in FGDs 2 and 4 were concerned with less efficacy in general and inappropriate storage conditions in particular. Members of the same FGDs had mixed thoughts and beliefs about the safety and effectiveness of the products purchased online. They seemed to have product authenticity issues and were doubtful about the promoted benefits of the products online. For example, a participant in FGD 2 mentioned that "*these products are promoted by creating a hype. This increases the sale of these items.*" A participant in FGD4 reported his experience whereby he purchased a fake product. Thereafter, he stated that *"all supplements purchased online are fake and the packaging does not represent what is included on the label."*

Further extension of theme 2 is discussed in detail based on the development of subthemes as below

**Subtheme 2(a): Efficacy issues with online purchases.**    Online products do create a negative impact and doubt in the minds of some consumers. A key reason is the inability to inspect products before purchasing and hence possible

disappointment when the item arrives. This can manifest as issues with quality or uncertainties regarding supply chain management. All these factors result in online products being perceived as less effective and thus not meeting the standard operation procedures for their manufacturing and storage.

*"Based on experiences and knowledge, products available online are less effective. There are doubts about the storage conditions too. Especially when medicines are considered as they must follow a standard operation procedure of storage and delivery to the consumers."* (P6 of FGD 2; P5 of FGD 4)

**Subtheme 2(b): Safety issues.** Compared to in-person purchases, online purchases tend to have efficacy and safety concerns. The propensity of fraud, data breaches, and the purchase of substandard or counterfeit goods is always there for online purchases. Therefore, efficiency is compromised through imprecise product information and difficulty in evaluating quality before purchase. The same concerns were also reported by the study participants.

*"In my views, supplement purchased online are forged. They do not have the active ingredient as mentioned on the labels. The packing does not represent what is included in the product and there is no method to check this too."* (P2 of FGD 2)

Participant 1 from FGD 2 shared that *"advertisement on food supplements for sports purposes create a perception that they are safe for all individuals. This is meant to create a publicity that a product is safe and hence increases the sale of these items".*

**Subtheme 2(c): The element of uncertainty.** Online customers are often faced with ambiguity. This may be due to the inability to physically inspect products, quality concerns and credibility of the sellers. In response, customers may require additional information, prefer familiar brands or insist on inspecting products at a physical venue. Such concerns were also reported by the current study respondents.

*"Uncertainty exists. Products purchased from online pharmacies/websites may not work like those purchased from pharmacies in person. There are chances of forgeries too. Sometimes it is better to visit and get products from a community pharmacy by us."* (P1 of FGD 2; P6 of FGD 4)

**Subtheme 2(d): Social proof and reviews.** Participants were doubtful about the products purchased from online pharmacies/websites. A doubt existed about the fake hype created by the sellers to increase their business (especially in the case of supplements). This doubt created reluctance for some buyers and was expressed during the discussion.

*"It is known that the reviews provided at the websites are fake or self-written. Suppliers create trust by faking positive reviews or in some cases negative reviews for their competitors. Under such circumstances, legitimacy of a product is always doubtful."* (P7 in FGD 4)

**Theme 3: Concerns regarding online purchase of medicine and supplements.** Participants expressed their experiences with online purchases and identified several obstacles in the process of buying medicines and health-related products online. Mixed feelings were observed whereby most of them expressed apprehensions regarding the leakage of personal information and the influx of unwanted advertisements received via email due to inadequate security and breaches of personal data. Furthermore, a lack of robust online security measures was seen as a significant concern, affecting participants' willingness to purchase medicines and health products online. Finally, the absence of clear refund or return policies among online vendors was another major deterrent. These factors collectively influenced participants' trust and interest in purchasing medicines and supplements from online pharmacies and websites.

***Subtheme 3(a): Leakage of personal information.*** This is a universal problem faced by online buyers. There is always a risk of personal information leakage that is later used for deception purposes. Almost all participants were concerned about this matter and perceived it as a major downside of buying from online platforms.

*"We as a customer give details of our financial institution. Starting from the bank to the cards and personal information. We have many examples where such information was used illegally. Unfortunately, it is still happening, and nobody takes the blame if some mishaps occur to the purchaser." (P1, P4 of FGD2; P3 of FGD 3; P5 of FGD 4)*

**Subtheme 3(b): Spam emails and unwanted advertisements.** Spam emails and unwanted advertisements can be very irritating, especially during an online transaction. In addition to wasting time, these can be malicious at times. Although there are several ways to minimize their impact (spam filters, etc.), such measures are the least employed by online vendors in developing countries. Frustration with spam emails and advertisements was again a major concern of the study respondents.

*"An irritating aspect of online shopping is receiving unwanted, spam emails with vulgar advertisements of banned medicines for recreational purposes and that is intolerable." (P5 of FGD1; P3 of FGD 4)*

Respondent 5 of FGD 3 showed an additional concern. She stated that "*at times, the advertisements are not morally and ethically accepted. It is quite shameful that such an advertisement pops up and becomes a matter of disgrace especially when you are sitting with someone.*"

**Subtheme 3 (c): Unregulated markets and lack of policies.** The cyber market is gigantic. With open windows, there are both regulated and unregulated vendors available. Such unregulated vendors and policies are reported to result in several issues, including the sale of counterfeit goods, scams, and the potential for financial losses. Often when such issues are reported, consumers face difficulties in seeking solutions due to the lack of clear legal frameworks.

*"The online market is huge, with legitimate and illegitimate vendors. However, there is no available measure to identify the legal vendors. Chances to purchase accidentally from unscrupulous or unregulated sellers are always there." (P4 in FGD 1)*

One participant shared his personal experience: *"I received a counterfeit product and not the one that I ordered. However, my complaint was never entertained by the seller and regrettably I was not aware of any platform for complaint."*

**Subtheme 3(d): Return and refund policy.** Online return and refund policies are complicated. Among those, the timeframe for returns, condition of items, shipping costs, proof of purchase, exceptions and exclusions are a few issues that are commonly faced by buyers. Even with all this evidence, the return and refund policies are intricate and take ages. These issues negatively impact consumers, highlighting the need for efficient and customer-friendly return systems.

*"Some online sellers may have unclear or unsatisfactory return and refund policies, which may raise concerns about the ease of returning or exchanging products in case of issues. This is a major obstacle in purchasing especially when the product is costly." (P7 of FGD 3; P5 of FGD 4)*

## Discussion

This study offers a comprehensive perspective on the online purchase of medicines and supplements, exploring the psychological factors involved and the concerns associated with the process. Over the past two decades, the online sale of medicines and supplements has grown significantly, gaining global adoption [18,58]. While it provides various benefits,

it also presents risks, particularly regarding patient safety [2,5,59,60]. Currently, in Pakistan there is limited data regarding the online purchase of medicines and supplements – the reasons, perceptions, and attitudes about obtaining them via the internet. This study helps fill that gap.

We observed that the participants were involved in the online purchasing of medicines and health supplements. There are an increasing number of individuals who are interested in purchasing medicine online, particularly those who are older and have higher education levels [19,61]. Older people prefer to purchase from online sources, as it provides improved access to and delivery of medicines, especially in remote areas [62]. Furthermore, people with higher education often have higher health literacy, which may help them access the internet more easily [19,63]. As reflected in our study, in which the majority of the participants are educated, purchasing medicine and/or health-related products online is easily accessible and convenient. Specifically, participants cited the ease of payment, convenience, low cost, choice, accessibility, and discounts as reasons for favoring the online purchase of medicines and health/food supplements. Furthermore, online purchase provides an alternative, low-cost option for health-related products and medicines. Due to convenience and doorstep delivery of medicines to consumers, there has been a rising global trend in e-pharmacy/online pharmacy purchases [64–68]. Convenience, accessibility, lower costs [69], and the ability to purchase drugs such as opioids and benzodiazepines for abuse are among the major reasons for online drug purchases [8].

Another reason for our study participants' preference for online sources/pharmacies is the availability of short items. Medicine shortages in physical stores can frustrate consumers as they might struggle to obtain the medicines they need, which may encourage them to seek alternative sources including the internet [70]. Furthermore, customer satisfaction with online purchases of medicines is reported to be higher, particularly regarding prices and services, compared to those offered by community pharmacies [11]. That being said, illegal online stores pose a serious hazard to global public health as they provide cheap, counterfeit medications that can severely impact morbidity and mortality [71].

Doubts about the efficacy of medicine, safety concerns, and uncertainty regarding storage conditions were the psychological factors that study participants mentioned, even though they had positive experiences making online purchases of medicines and health-related items. Fear regarding counterfeit medicine from online medicine sellers is another aspect that might impact a consumer's decision to acquire medicine online or not. Trust is an essential factor in customers' purchase behavior in the context of e-commerce. An increasing number of counterfeit products are sold via the internet. These negatively impact individuals' health and call for a fight against their sale [72]. The availability of this falsified and spurious medicine is due to poor regulatory systems and the non-implementation of drug laws [73].

Online reviews also play an important role in shaping consumers' decisions whether to buy medicines through the internet. The creation of fake positive reviews by illegal medicine sellers may increase customer preferences for purchasing from them by artificially boosting their website ratings and reputation, making their platforms appear more trustworthy and credible [70].

The participants in our study expressed mixed feelings regarding the purchase of medicine and health supplements from online pharmacies/sources, particularly regarding the risk of personal information leakage and the influx of unwanted advertisements via email, which they attributed to inadequate online security measures. In traditional brick-and-mortar pharmacies, the need to protect the privacy and confidentiality of clients and patients has been taken into account when designing premises and customer service procedures [74,75]. Even though confidentiality and privacy are key elements, these are not ensured in online pharmacies and websites due to third party analytics [76–80]. Furthermore, spam and unsolicited commercial email is a problem for every person in the world that uses the internet [81]. Illegal spam and unwanted emails normally leads a consumer to a harmful site. Email spam, malware/spyware and other cybersecurity threats tend to be used for financial fraud and data phishing activities [13]. As reported, those consumers who visited online sources for purchases receive adult-related emails advertising sexual health medication and pornographic items [81].

Regarding the return/refund of medication or health supplements, the participants in our study reported this as one of their concerns when purchasing from online sources. However, a study also reported that 50% of the participants who

purchased medicines from online pharmacies expressed satisfaction with the exchange and return processes when receiving counterfeit or ingenuine products [10]. In Pakistan, the unregulated online medicine business underlines the need for proper regulation and compliance with the rules in order to safeguard the precious lives of consumers. A proper regulated online business model for medication delivery will contribute significantly to the economic stability of Pakistan in this hard time. The need to regulate and acknowledge the importance of pharmaceutical professionals in online pharmacies and websites is vital to counter falsified, counterfeit medicines that are sold online.

### Limitations of the study

As with other studies, this qualitative study is not without limitations. We acknowledge that FGDs are vulnerable to group-level biases. Although the research team established clear ground rules and the moderator actively encouraged balanced participation, there remained a tendency for dominant participants to overshadow the discussions. Moreover, the short duration of the FGDs may have been a barrier to participants providing personal narratives. While the application of grounded theory provided a systematic approach to data analysis, it has the tendency toward researchers' interpretation and probable bias. Lastly, the use of convenience sampling and the urban setting restricted the generalizability of findings to other regions or socio-economic groups in Pakistan.

To address these limitations, we recommend using follow-up interviews, and integrating supplementary activities to enhance narrative depth. Ensuring diverse participant representation and using stratified sampling would enhance generalizability. Triangulation of data sources and involving multiple analysts can help reduce researcher bias, while professional translation services can ensure linguistic accuracy.

### Practical implications

This study has several promising implications. Logically, the study findings highlight the need for consumer education on safe online purchasing. Parallel to this, online pharmacies are encouraged to improve transparency, return policies, and data protection. Notionally, where the study findings contributed to understanding consumer behavior toward digital health services, methodologically it demonstrates the value of using qualitative methods in future research. From a policy perspective, the findings call for stronger regulation of online pharmacies, including vendor certification, prescription enforcement, and consumer data protection.

### Future work

This study gives a thorough examination of the many motivations that lead consumers to purchase prescription medications over the internet. Identifying those factors might offer the foundation for authorities to create evidence-based awareness efforts to reduce the use of the internet to obtain prescription medications. Furthermore, this review provides future research recommendations to improve existing knowledge and solve research gaps in this field.

### Conclusion

Acquiring medication from online pharmacies not only provides customers with ease and convenience but also presents them with a diverse range of options for purchasing medicines and supplements. Nevertheless, it is essential for regulatory authorities to strictly oversee the online pharmacy market to ensure the safe and reliable availability of medicines, thereby addressing concerns through proper regulation. Effective governance of online pharmacies would not only improve consumer trust and safety but also strengthen the e-commerce sector, thereby contributing to Pakistan's economic growth.

### Supporting information

**S1 File. Inclusivity in global research.**
(DOCX)

## Author contributions

**Conceptualization:** Amal K Suleiman, Ateeq Ur Rehman.

**Data curation:** Ateeq Ur Rehman.

**Formal analysis:** Amal K Suleiman, Abbas Albarq, Ateeq Ur Rehman.

**Funding acquisition:** Amal K Suleiman.

**Investigation:** Amal K Suleiman, Ateeq Ur Rehman.

**Methodology:** Amal K Suleiman, Abbas Albarq, Ateeq Ur Rehman.

**Project administration:** Ateeq Ur Rehman.

**Writing – original draft:** Ateeq Ur Rehman.

**Writing – review & editing:** Amal K Suleiman, Abbas Albarq.

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
