## [Decision Letter · Decision Letter 0]

4 Jun 2025

Dear Dr. Rehman,

Thank you for submitting your manuscript to PLOS ONE. After careful consideration, we feel that it has merit but does not fully meet PLOS ONE’s publication criteria as it currently stands. Therefore, we invite you to submit a revised version of the manuscript that addresses the points raised during the review process.

We look forward to receiving your revised manuscript.

Kind regards,

Jenny Wilkinson, PhD

Academic Editor

PLOS ONE

Journal Requirements:

Reviewers' comments:

Reviewer's Responses to Questions

**Comments to the Author**

1. Is the manuscript technically sound, and do the data support the conclusions?

Reviewer #1: Partly

Reviewer #2: Yes

2. Has the statistical analysis been performed appropriately and rigorously?

Reviewer #1: No

Reviewer #2: N/A

3. Have the authors made all data underlying the findings in their manuscript fully available?

Reviewer #1: No

Reviewer #2: Yes

4. Is the manuscript presented in an intelligible fashion and written in standard English?

Reviewer #1: Yes

Reviewer #2: Yes

Reviewer #1: Dear Authors,

Please see my comments below:

- Title: The title does not reflect the geographical focus and research methodology.

- Abstract: The novel findings of the research are not reflected in the abstract. What exactly does this study add to existing knowledge? Please make more explicit statements. Additionally, consider updating the methods section in the abstract to help readers understand the context: when, where, and why was the study conducted?

- Introduction: The first paragraph is vague. As we are now in the post-COVID world, what are the implications of the last sentences of the first paragraph?

- Introduction, lines 96-103: The introduction should place the entire topic in context. What is the regulatory framework for online pharmacies and distance sales of pharmaceutical products in Pakistan? Are there verified online pharmacies in the country? What is already known about consumer attitudes towards e-pharmacies? Why are you conducting your research now?

- Methods, general: How have you incorporated previously published survey instruments into your study and questions? Numerous research papers have described attitudes, motivations, risk perceptions, etc., regarding online pharmacies. How do you build on these, and what exactly does your research add to existing knowledge?

- Methods, lines 107-109: The product category focus is not clear. Here, you mention “purchasing behaviors and experiences with buying food supplements for sports, health supplements, and medicines from online sources.” Participant inclusion refers to acquiring dietary supplements for sports, health supplements, or medication from an online/Internet source. Accordingly, what if consumers have not purchased any pharmaceutical product from an online pharmacy, but rather dietary supplements from a webshop?

- Methods, line 150: Please be more specific. What exactly does “The study was approved by the ethics committee of the respective university” mean? Which university, and what was the approval process?

- Results, line 233: What do you mean by “Category 1.4: Availability of market short items”? Does this refer to products affected by shortages?

-Results, lines 258-271: Categories 2.1, 2.2, and 2.3 seem to indicate similar, if not the same, issues. Please describe the differences and similarities.

- Results, theme 3, line 304: How do you know the products purchased were substandard? Please refer to WHO terminology on substandard and falsified (SF) medicinal products.

- General limitation: I see the sample size and demographic parameters of the respondents. In the discussion, you mention: “participants in this study, who were majority university graduates.” How does your study represent the national population?

Reviewer #2: Dear authors, Good job on your manuscript.

However some modifications may be necessary to improve the quality of the manuscript.

Generally, the article requires proof reading for language quality. For instance some grammatical errors identified are:

1. Remove the 2019 before COVID 19 in line 52

2. Replace content with "Food and healthcare supplements" in line 109

3. Replace for with to in line 115

4. Remove above in line 120.

5. Rephrase "and reasons for online purchases" in lines 123-124

6. In line 128 replace with "the participants were recruited after written informed consent was obtained"

7. Remove more in line 131

8. Change in-person to in-store in line 137

9. Rewrite lines 340-342 for clarity.

In addition to grammatical issues, some structural adjustments can also be made. For instance authors could:

1. Expand the last paragraph of the introduction to include (a) potential of online pharmacies in Pakistan's general population, pharmacy outlets and health sector, (b) nature of available policies/ regulations on sales of medicine and supplements (c) the nature of available studies conducted on online pharmacies in Pakistan, the inability of these studies to answer the current studies research question, and the need for an in-depth understanding of consumers concerns subjectively using qualitative approach. (d) Summarize introductory section with the potential benefits of conducting the study and organization of its sections.

2. A two paragraph literature review summarizing studies on drivers and barriers of e-pharmacy use globally will benefit the manuscript readers.

3. Write more detailed explanation of subthemes 1.1, 1.2, 1.3, 2.1, 2.2, 2.3, 2.4, 3.1 -3.6. Readers rely heavily on the authors the interpretation of transcripts content so these sections need to be rich. Conflicting statements by participants in focus group discussion could also furnish interpretations.

4.Include practical implications of the study's findings for consumers, online pharmacy vendors, research community and policy makers.

5. References font style is not consistent with text.

**Do you want your identity to be public for this peer review?** For information about this choice, including consent withdrawal, please see our Privacy Policy

Reviewer #1: No

Reviewer #2: **Yes: ** Adetumilara Iyanuoluwa ADEBO

---

## [Author Response · Author response to Decision Letter 1]

4 Jul 2025

Dear Editor and Reviewers,

Thank you for taking the time to provide such detailed and helpful comments on our manuscript. We appreciate your constructive criticisms and suggestions, and we believe they will significantly improve the quality of our manuscript. We have revised the manuscript based on your feedback and would like to provide the following responses to your concerns.

Reviewer #1

Please see my comments below:

Comment: Title: The title does not reflect the geographical focus and research methodology.

Response: Thank you for comment, the tile is modified to “A qualitative exploration of consumer’s views and experiences towards online pharmacies: narratives from a developing world”.

Comment: Abstract: The novel findings of the research are not reflected in the abstract. What exactly does this study add to existing knowledge? Please make more explicit statements. Additionally, consider updating the methods section in the abstract to help readers understand the context: when, where, and why was the study conducted?

Response: Thank you for comment, the abstract is rewritten again as suggested.

Comment: Introduction: The first paragraph is vague. As we are now in the post-COVID world, what are the implications of the last sentences of the first paragraph?

Response: Thank you for comment, the introduction is rewritten again as suggested.

Comment: Introduction, lines 96-103: The introduction should place the entire topic in context. What is the regulatory framework for online pharmacies and distance sales of pharmaceutical products in Pakistan? Are there verified online pharmacies in the country? What is already known about consumer attitudes towards e-pharmacies? Why are you conducting your research now?

Response: Thank you for comment, it is rewritten again and the justification/rational of the study is also updated to avoid confusion to the readers.

Comment: Methods, general: How have you incorporated previously published survey instruments into your study and questions? Numerous research papers have described attitudes, motivations, risk perceptions, etc., regarding online pharmacies. How do you build on these, and what exactly does your research add to existing knowledge?

Response: Thank you for comment, the authors appraise the comment and concern. Although we have added the details in the revised manuscript, it is important to explain the comment in detail. An in-depth qualitative analysis that was conducted due to lack of evidence from a developing country (Pakistan). Therefore, it was obvious to take assistance from studies of the same nature. The authors identified relevant and validated tolls and guide through an extensive literature review. The established protocols were followed, the original sources were cited, and piloting was conducted before adapting the tools, guide (whole or in parts). The pilot phase ensured culture sensitivity, reliability, validity and integration of the developed guide for the field study.

The current study does add to the existing literature. Being the pioneer study from an urban area of Pakistan, the study reveals certain limitations and trepidations. This inevitably shows that if urban population has unanswered concerns and hesitations, the vast population living in rural areas will have unresolved concerns and that calls for a larger mishap when it comes to online purchase. These concerns are genuine and must be addressed at the earliest to foster online business in a legal manner in developing countries.

Comment: Methods, lines 107-109: The product category focus is not clear. Here, you mention “purchasing behaviors and experiences with buying food supplements for sports, health supplements, and medicines from online sources.” Participant inclusion refers to acquiring dietary supplements for sports, health supplements, or medication from an online/Internet source. Accordingly, what if consumers have not purchased any pharmaceutical product from an online pharmacy, but rather dietary supplements from a webshop?

Response: Thank you for comment, we appreciate the comment. We do agree that the initial transcription was classical and was complex. Based on the comments, we have re-transcribed and with mutual consent have revised as suggested. The inclusion criteria are also rephrased so make it more understanding in nature. Please refer to the revised manuscript.

Furthermore, we have added operational definition of terminologies that will make things easier for the readers. Lastly, we confirmed from the respondents about the purchase inventory, and it was assured that they purchased pharmaceuticals as well as health products in random.

Comment: Methods, line 150: Please be more specific. What exactly does “The study was approved by the ethics committee of the respective university” mean? Which university, and what was the approval process?

Response: Thank you for comment, we have rephrased the statement. “Institutional Ethical Committee of the Foundation University, Rawalpindi Campus, Pakistan approved the study (EC/FUI/2023/32). Written consent of participation and publication was taken from the respondents before the FGD. The participants were introduced to the nature of the research before the beginning of the sessions, were made secure of the confidentiality of their responses and their right to withdraw from the study.”

Comment: Results, line 233: What do you mean by “Category 1.4: Availability of market short items”? Does this refer to products affected by shortages?

Response: Thank you for comment, the initial version of the manuscript used the word category for subthemes. However, while revising the manuscript and going through the reviews, we felt that it is better to stick to the traditional phrases of qualitative studies (Themes and Subthemes).

We really appreciate the reviewer for such attentions to details in the review report. The short items in the Pakistani context are referred to the products that are often unavailable or when available are on high prices. Please refer to the operational definitions and examples.

Comment: Results, lines 258-271: Categories 2.1, 2.2, and 2.3 seem to indicate similar, if not the same, issues. Please describe the differences and similarities.

Response: Thank you for comment, yes, after re-transcription we do agree that there were some similarities. Although honest, they may be confusing for the readers. We have rephrased and the changes can be referred to the revised manuscript.

Comment: Results, theme 3, line 304: How do you know the products purchased were substandard? Please refer to WHO terminology on substandard and falsified (SF) medicinal products.

Response: Thank you for comment, yes, we did refer to the WHO definition. The product being substandard or inferior was a concern shown by the respondents and we did not essentially think to establish the control of products as it was not part of the objective.

Comment: General limitation: I see the sample size and demographic parameters of the respondents. In the discussion, you mention: “participants in this study, who were majority university graduates.” How does your study represent the national population?

Response: Thank you for comment, the text “participants in this study, who were majority university graduates” has been replaced with “educated” to avoid confusion. By the term university graduates we meant graduated with a degree, but now replaced to avoid confusion to the readers, additionally the discussion rewritten again. We have added a separate header as “limitations.”

Reviewer #2

Dear authors, good job on your manuscript.

However some modifications may be necessary to improve the quality of the manuscript.

Generally, the article requires proof reading for language quality. For instance, some grammatical errors identified are:

1. Remove the 2019 before COVID 19 in line 52

Response: changed as suggested.

2. Replace content with "Food and healthcare supplements" in line 109

Response: changed to health supplements as suggested.

3. Replace for with to in line 115

Response: changed as suggested.

4. Remove above in line 120.

Response: changed as suggested.

5. Rephrase "and reasons for online purchases" in lines 123-124

Response: changed as suggested.

6. In line 128 replace with "the participants were recruited after written informed consent was obtained"

Response: changed as suggested.

7. Remove more in line 131

Response: changed as suggested.

8. Change in-person to in-store in line 137

Response: changed as suggested.

In addition to grammatical issues, some structural adjustments can also be made. For instance, authors could:

1. Expand the last paragraph of the introduction to include (a) potential of online pharmacies in Pakistan's general population, pharmacy outlets and health sector, (b) nature of available policies/ regulations on sales of medicine and supplements (c) the nature of available studies conducted on online pharmacies in Pakistan, the inability of these studies to answer the current studies research question, and the need for an in-depth understanding of consumers concerns subjectively using qualitative approach. (d) Summarize introductory section with the potential benefits of conducting the study and organization of its sections.

Response: Thank you for comment, the introduction section is rewritten again as suggested.

2. A two paragraph literature review summarizing studies on drivers and barriers of e-pharmacy use globally will benefit the manuscript readers.

Response: Thank you for comment, the manuscript is rewritten again as suggested to benefit the readers.

3. Write more detailed explanation of subthemes 1.1, 1.2, 1.3, 2.1, 2.2, 2.3, 2.4, 3.1 -3.6. Readers rely heavily on the authors the interpretation of transcripts content so these sections need to be rich. Conflicting statements by participants in focus group discussion could also furnish interpretations.

Response: Thank you for comment, we appreciate the comment. We have added details as required. Please refer to the result section of the revised manuscript. The entire section is rephrased.

4.Include practical implications of the study's findings for consumers, online pharmacy vendors, research community and policy makers.

Response: Thank you for comment, the section on limitation and practical implications of the study is added to the manuscript as suggested.

5. References font style is not consistent with text.

Response: Thank you for comment, the suggested changes are done in the manuscript.

Thank you for your time and valuable feedback. We believe that the revisions have significantly improved the manuscript and hope that it is now suitable for publication.

Regard’s

Corresponding author

---

## [Decision Letter · Decision Letter 1]

29 Jul 2025

Dear Dr. Rehman,

Thank you for submitting your manuscript to PLOS ONE. After careful consideration, we feel that it has merit but does not fully meet PLOS ONE’s publication criteria as it currently stands. Therefore, we invite you to submit a revised version of the manuscript that addresses the points raised during the review process.

We look forward to receiving your revised manuscript.

Kind regards,

Jenny Wilkinson, PhD

Academic Editor

PLOS ONE

Journal Requirements:

Reviewers' comments:

Reviewer's Responses to Questions

**Comments to the Author**

Reviewer #2: All comments have been addressed

2. Is the manuscript technically sound, and do the data support the conclusions?

Reviewer #2: Yes

3. Has the statistical analysis been performed appropriately and rigorously?

Reviewer #2: N/A

4. Have the authors made all data underlying the findings in their manuscript fully available?

Reviewer #2: Yes

5. Is the manuscript presented in an intelligible fashion and written in standard English?

Reviewer #2: Yes

Reviewer #2: Dear Authors,

Good job on making improvements to your manuscript. However, some minor and major adjustments are required to improve the quality and clarity of the writing. The following sections describe the required revisions.

Minor revisions

The manuscript requires extensive proofreading for grammatical accuracy.

1. Include "increase" in line 71.

2. Add "in" before Africa, Asia in line 106. Remove "India" from the statement, as India is part of Asia. Add "regions" after Latin America.

3. Add "es" after business in line 111. Remove "s" after strive.

4. Either remove "from" in line 451 or add "sources" after online.

5. Rewrite the sentence that begins in line 461 for clarity.

6. Rewrite the sentence that begins in line 468 and ends in line 470 for clarity.

7. Add a sentence or two in line 456 justifying why older and more educated people tend to prefer e-pharmacies. You can copy and rephrase from lines 93-96, but ensure that there is a statement on the education side of the argument.

8. "Preference for" is more appropriate than "diversion towards" in line 465, as people could still use physical stores while preferring e-pharmacies.

9. Medicine shortage "in physical stores" in line 466.

10. The sentence beginning in line 471 is not necessary where it is. Authors should either expunge or insert it in a more appropriate section.

11. The sentence beginning in line 491 can also be expunged as the paragraph could end without it.

12. Incomplete word "interne" in line 502.

Replace 'involved in' with "used for" in line 504.

13. Replace "of" in line 511 with "for".

14. Replace "life" with "lives" in line 512.

15. Replace "safe" with "safeguard" in line 521

Major revisions

1. Contents in the limitation section do not appear like limitations. What are the actual limitations of the research? For instance, are there biases relating to group dynamics during data collection, e.g., dominant voices, social desirability bias, shallow personal narratives? Are there any limitations based on transcription and/or analysis? Are there any limitations that may arise using grounded theory, i.e., challenges with achieving theoretical saturation or researcher bias?

2. The practical implications section should be rewritten, as there are no specific practical implications mentioned. Also, the authors should specify and separate practical implications from theoretical, methodological, and policy implications

**Do you want your identity to be public for this peer review?** For information about this choice, including consent withdrawal, please see our Privacy Policy

Reviewer #2: **Yes: ** Adetumilara Adebo

---

## [Author Response · Author response to Decision Letter 2]

6 Aug 2025

Thank you for taking the time to provide such detailed and helpful comments on our manuscript. We appreciate your constructive criticisms and suggestions, and we believe they will significantly improve the quality of our manuscript. We have revised the manuscript based on your feedback and would like to provide the following responses to your concerns.

Reviewer #2

Dear Authors,

Good job on making improvements to your manuscript. However, some minor and major adjustments are required to improve the quality and clarity of the writing. The following sections describe the required revisions.

Minor revisions

The manuscript requires extensive proofreading for grammatical accuracy.

1. Include "increase" in line 71.

2. Add "in" before Africa, Asia in line 106. Remove "India" from the statement, as India is part of Asia. Add "regions" after Latin America.

3. Add "es" after business in line 111. Remove "s" after strive.

4. Either remove "from" in line 451 or add "sources" after online.

5. Rewrite the sentence that begins in line 461 for clarity.

6. Rewrite the sentence that begins in line 468 and ends in line 470 for clarity.

7. Add a sentence or two in line 456 justifying why older and more educated people tend to prefer e-pharmacies. You can copy and rephrase from lines 93-96, but ensure that there is a statement on the education side of the argument.

8. "Preference for" is more appropriate than "diversion towards" in line 465, as people could still use physical stores while preferring e-pharmacies.

9. Medicine shortage "in physical stores" in line 466.

10. The sentence beginning in line 471 is not necessary where it is. Authors should either expunge or insert it in a more appropriate section.

11. The sentence beginning in line 491 can also be expunged as the paragraph could end without it.

12. Incomplete word "interne" in line 502.

Replace 'involved in' with "used for" in line 504.

13. Replace "of" in line 511 with "for".

14. Replace "life" with "lives" in line 512.

15. Replace "safe" with "safeguard" in line 521

Response: Thank you for comments and highlighting typo/grammatical errors. The manuscript is editing for English language by native English speaker to avoid any errors and confusion for the readers. Additionally, text on the older and educated people tends to prefer e-pharmacies is also added to the manuscript as suggested.

Major revisions

Comment: Contents in the limitation section do not appear like limitations. What are the actual limitations of the research? For instance, are there biases relating to group dynamics during data collection, e.g., dominant voices, social desirability bias, shallow personal narratives? Are there any limitations based on transcription and/or analysis? Are there any limitations that may arise using grounded theory, i.e., challenges with achieving theoretical saturation or researcher bias?

Response: Thank you for comments, the limitation section is rewritten again as per suggestions.

Comment: The practical implications section should be rewritten, as there are no specific practical implications mentioned. Also, the authors should specify and separate practical implications from theoretical, methodological, and policy implications.

Response: Thank you for comments, the practical implications section is rewritten again as per suggestions.

Thank you for your time and valuable feedback. We believe that the revisions have significantly improved the manuscript and hope that it is now suitable for publication.

Regard’s

Corresponding author

---

## [Editor Report · Decision Letter 2]

8 Aug 2025

Dear Dr. Rehman,

Thank you for submitting your manuscript to PLOS ONE. After careful consideration, we feel that it has merit but does not fully meet PLOS ONE’s publication criteria as it currently stands. Therefore, we invite you to submit a revised version of the manuscript that addresses the points raised during the review process.

We look forward to receiving your revised manuscript.

Kind regards,

Jenny Wilkinson, PhD

Academic Editor

PLOS ONE

Journal Requirements:

Additional Editor Comments:

Thank you for your responses, these have satisfactorily addressed the comments. I note that some of the participant quotes have been edited, as these are quotes the wording should be as provided by respondents, even if not grammatically correct and modifications should indicated. Please confirm that the quotes are as provided by respondents.

---

## [Author Response · Author response to Decision Letter 3]

9 Aug 2025

Dear Editor,

Thank you for taking the time to provide such detailed and helpful comments on our manuscript. We appreciate your constructive criticisms and suggestions, and we believe they will significantly improve the quality of our manuscript. We have revised the manuscript based on your feedback and would like to provide the following responses to your concerns.

In previous revision I got comments, and I addressed the comments as below:

Comment from Editor:

Comment 1: Thank you for your responses; these have satisfactorily addressed the comments. Response: We appreciate your feedback and are pleased that our previous responses have addressed the earlier concerns.

Comment 2: I note that some of the participant quotes have been edited.

Response: We confirm that the quotes presented in the manuscript reflect the original responses provided by the participants, we did English editing for the that but as suggested in the comment below were reverted to its original quotes/wordings and are mentioned in the result section of manuscript.

Comment 3: As these are quotes, the wording should be provided by respondents, even if not grammatically correct.

Response: While minor grammatical and language edits were initially applied to enhance readability, these have now been removed in accordance with your recommendation.

Comment 4: Modifications should be indicated.

Response: No modifications remain in the current version. All quotes are now presented as quotes provided by the respondents without making any changes.

However, in additional comments, the responses to these are below:

Comment 1: Please amend your Response to Reviewers letter to include a point-by-point response to each of the points made by the Editor and / or Reviewers. Please follow this link for more information: http://blogs.PLOS.org/everyone/2011/05/10/how-to-submit-your-revised-manuscript/

Response: Thank you for comment, the rebuttal/response to comments by editor are done point to point in above text as suggested. Previously the response was in form of a paragraph but now it is point-by-point as per editors’ recommendations.

Comment 2: Please amend the title either on the online submission form or in your so that they are identical.

Response: Thank you for commenting and highlighting this issue, the title has been amended and made uniform in both the manuscript file and the editorial manager.

Thank you for your time and valuable feedback. We believe that the revisions have significantly improved the manuscript and hope that it is now suitable for publication.

Regard’s

Corresponding author

---

## [Editor Report · Decision Letter 3]

13 Aug 2025

A qualitative exploration of consumers' views and experiences toward online pharmacies: narrative from a developing country

PONE-D-25-09748R3

Dear Dr. Rehman,

We’re pleased to inform you that your manuscript has been judged scientifically suitable for publication and will be formally accepted for publication once it meets all outstanding technical requirements.

Kind regards,

Jenny Wilkinson, PhD

Academic Editor

PLOS ONE
---

## [Editor Report · Acceptance letter]

PONE-D-25-09748R3

PLOS ONE

Dear Dr. Rehman,

I'm pleased to inform you that your manuscript has been deemed suitable for publication in PLOS ONE. Congratulations! Your manuscript is now being handed over to our production team.

Kind regards,

on behalf of

Dr Jenny Wilkinson

Academic Editor

PLOS ONE